

# Non-destructive insect metabarcoding for surveillance and biosecurity in citrus orchards: recording the good, the bad and the psyllids

Francesco Martoni[1], Reannon Smith[1], Alexander M. Piper[1], Jessica Lye[2], Conrad Trollip[1], Brendan C. Rodoni[1,3] and Mark J. Blacket[1]

[1] Agriculture Victoria Research, State Government Victoria, Bundoora, Victoria, Australia
[2] Citrus Australia Ltd., Wandin North, Victoria, Australia
[3] School of Applied Systems Biology, La Trobe University, Bundoora, Victoria, Australia

## ABSTRACT

**Background:** The Australian citrus industry remains one of the few in the world to be unaffected by the African and the Asian citrus psyllids, *Trioza erytreae* Del Guercio and *Diaphorina citri* Kuwayama, respectively, and the diseases their vectored bacteria can cause. Surveillance, early detection, and strict quarantine measures are therefore fundamental to safeguard Australian citrus. However, long-term targeted surveillance for exotic citrus pests can be a time-consuming and expensive activity, often relying on manually screening large numbers of trap samples and morphological identification of specimens, which requires a high level of taxonomic knowledge.

**Methods:** Here we evaluated the use of non-destructive insect metabarcoding for exotic pest surveillance in citrus orchards. We conducted an 11-week field trial, between the months of December and February, at a horticultural research farm (SuniTAFE Smart Farm) in the Northwest of Victoria, Australia, and processed more than 250 samples collected from three types of invertebrate traps across four sites.

**Results:** The whole-community metabarcoding data enabled comparisons between different trapping methods, demonstrated the spatial variation of insect diversity across the same orchard, and highlighted how comprehensive assessment of insect biodiversity requires use of multiple complimentary trapping methods. In addition to revealing the diversity of native psyllid species in citrus orchards, the non-targeted metabarcoding approach identified a diversity of other pest and beneficial insects and arachnids within the trap bycatch, and recorded the presence of the triozid *Casuarinicola* cf *warrigalensis* for the first time in Victoria. Ultimately, this work highlights how a non-targeted surveillance approach for insect monitoring coupled with non-destructive DNA metabarcoding can provide accurate and high-throughput species identification for biosecurity and biodiversity monitoring.

Corresponding author
Francesco Martoni,
francesco.martoni@agriculture.vic.gov.au

## INTRODUCTION

Rapid diversification in the late Miocene, along with natural range expansion from South-East Asia and selective breeding programs in more recent times, has contributed to the hundreds of citrus species and hybrids in existence today (*Wu et al., 2018*; *Goh et al., 2022*). Citrus is grown commercially in more than 140 countries worldwide, and its 2021 production (including oranges, lemons & limes, tangerines, mandarins, clementines, pomelos & grapefruits, and other citrus fruits) was greater than 160 million tons (*FAOSTAT, 2023*).

Citrus greening, also known as Huanglongbing (HLB), is considered to be the most serious disease for citrus industries worldwide (*Bové, 2006*). This disease significantly impacts fruit production, resulting in green, misshapen, and bitter fruits, which cannot be sold fresh or used for juice (*Dala-Paula et al., 2019*). This disease has been known in Asia for more than a century and has spread to many important global citrus producing regions in the last 20 years (*Coletta-Filho et al., 2004*; *Halbert, 2005*; *Batool et al., 2007*). Huanglongbing is associated with three non-cultivable phloem-restricted bacteria *Candidatus* Liberibacter asiaticus (CLas), *Ca.* L. africanus (CLaf) and *Ca.* L. americanus (CLam). These are vectored by the Asian citrus psyllid (ACP), *Diaphorina citri* Kuwayama (Psyllidae), and the African citrus psyllid (AfCP) *Trioza erytreae* Del Guercio (Triozidae), with ACP transmitting CLas and CLam, and AfCP transmitting CLaf and, under laboratory conditions, CLas (*Lallemand, Fos & Bové, 1986*; *Reynaud et al., 2022*). CLas is now widespread to more than 50 countries worldwide, including temperate, tropical, and subtropical areas, where it always occurs associated with its psyllid vector (*EPPO, 2023*; *Wang, 2022*).

A number of aspects of the disease contribute to its economic importance, such as the difficulties in early diagnosis, challenges in controlling the rapidly spreading insect vectors, the lack of effective treatment, and elevated producer costs associated with managing the disease (*Snyder et al., 2022*; *FAO, 2022*). In the United States, since the disease was detected in 2005 (*Bové, 2006*), citrus production in Florida has been significantly impacted. Once the predominant citrus producer in the United States, Florida's 2020/21 production reduced to a third of levels of 2005/06 (*USDA, 2007*, *2021*). A concomitant reduction in the number of citrus groves, juice processing facilities, and packing houses indicates a dire situation for the Florida citrus industry with substantial economic losses being attributed to HLB (*Singerman, Burani-Arouca & Futch, 2018*; *Singerman & Rogers, 2020*; *USDA, 2021*).

In Australia, citrus is a significant horticultural crop with production made up of oranges, mandarins, lemons, limes, grapefruit and tangelos. In 2019/20, the national citrus crop was valued at AUD$942 million, with 37% being exported, 36% being sold domestically as a fresh product and 26% being processed into value added products (*Hogan et al., 2022*). In 2019 Australia was the world's tenth largest citrus exporter and the second largest exporter to South East Asian countries (*Hogan et al., 2022*), with total export valued at AUD$509 million: an increase of 171% since 2012/13 (*Hogan et al., 2022*). Encompassing parts of New South Wales and Victoria, the Sunraysia region is one of

Australia's most highly productive citrus growing regions, contributing to approximately 20% of the national citrus crop (*Agriculture Victoria, 2018*; Citrus Australia, 2022, personal communication).

Recognizing the threat posed by exotic pest species, particularly the ACP-HLB complex, and to better understand the psyllid diversity already present in the country, the Australian citrus industry has undergone a period of increased investment into exotic citrus pest surveillance and biosecurity activities (Citrus Australia, 2022, personal communication). While no native Australian psylloid is known to be a pest of citrus (*Hollis, 2004*), psylloids can be easily windblown onto plants other than their hosts (*Burckhardt et al., 2014*), often in high numbers, raising false alarms for growers (*Martoni & Blacket, 2021*).

The Australian citrus industry has recently started an early detection network using sticky traps, in addition to visual inspection, stem-tap sampling, and "budstick" sampling citrus in several urban and citrus growing regions each year, to complement surveillance undertaken by government agencies and research programs (Citrus Australia, 2022, personal communication). One benefit of these activities is the enhancement of citrus industry knowledge relating to existing psyllid species in Australia. However, current surveillance efforts involve a large resource investment as sticky traps deployed for early detection of Asian citrus psyllid require visual inspection and triaging by trained entomologists before suspect specimens are submitted for molecular diagnostics (Citrus Australia, 2022, personal communication). Sticky traps also present issues for large scale sampling, with specimens being difficult to remove from the trap, and unlike some alternative trapping methods do not include a preservative, allowing molecular and morphological characters to degrade under harsh field conditions, impeding identification.

Today, high-throughput sequencing (HTS) technologies, especially metabarcoding, are being evaluated and adopted worldwide for biosecurity surveillance of pests and pathogens (*Batovska et al., 2018*; *Tedersoo et al., 2019*; *Hardulak et al., 2020*; *Trollip et al., 2022*, *2023*; *Young et al., 2021*; *Martoni et al., 2023*). Metabarcoding allows DNA barcode-based identification to be conducted in a massively parallel manner, generating a large number of individual barcode sequences in a single reaction to enable the simultaneous identification of diverse insect species in large mixed samples (*Piper et al., 2019*; *Liu et al., 2020*). Therefore, metabarcoding offers a number of advantages compared to more traditional insect identification techniques, which are often limited by the cost of sorting and identifying many specimens and the high level of taxonomical expertise required for this (*Karlsson et al., 2018*). Indeed, metabarcoding has been recently used to assess the insect diversity across different orchards in China, to record both pests and beneficial insects (*Liu et al., 2023*).

Here, we deployed an iMapPESTS Sentinel smart trap (hereafter referred to as the Sentinel) in the SuniTAFE Smart Farm of Irymple, near Mildura (Victoria, Australia), during an 11-weeks trial. After previously evaluating the Sentinel in a grain growing region (*Martoni et al., 2023*), and comparing it with a similar suction trap, we tested it here in a multi-commodity horticultural farm, that included citrus orchards, comparing it against different insect collection methods across different sites. Insect samples from the Sentinel, ground traps and passive wind traps were collected into a propylene glycol solution to
preserve their DNA, then processed using a non-destructive insect metabarcoding technique (*Martoni et al., 2022*). This non-targeted diagnostic approach provides a comprehensive characterisation of the trapped insect diversity, encompassing both pests and beneficial arthropod species, such as parasitoids and predators.

The aims of this work were to: (i) assess the insect composition and diversity within the target agroecosystem, confirming the presence or absence of target psylloid species (Hemiptera: Psylloidea); (ii) compare this diversity between ground, wind, and suction traps to determine if the insect composition is similar using different trapping methods; (iii) compare the insect diversity across different sites within the same farm, to assess whether sampling at a single collection site is adequate for recording the overall diversity within a larger farm; (iv) explore the value of metabarcoding when used to observe the presence of different insect groups, including target and non-target pests and beneficials.

## MATERIALS AND METHODS

### Field trial and insect trapping

The field trial was run for 11 weeks from the 10th of December 2021 to 23rd of February 2022 at the SuniTAFE Smart Farm (34°15′57.6′′S 142°08′47.7′′E) in Mildura, Victoria, Australia (Fig. 1). The maps in Fig. 1 were generated using Google Earth v 7.3.6.9345 (Google, Mountain View, CA, USA). Insect collection at this facility required no permit. Four separate sites (1–4, Fig. 1) within the Smart Farm were selected for deployment of insect traps. These sites included a car park (site 1), a semi-grassy area between a citrus orchard and avocados (site 2), an orchard containing only citrus (site 3) and a strip of native vegetation (mainly *Acacia* spp. and *Eucalyptus* spp.; site 4). At each site, a passive wind trap (Fig. 1) and four ground traps were deployed. The ground traps consisted of a plastic box (18 cm × 12 cm × 6.5 cm) filled with ~250–400 ml of preservative (see below). These boxes were deployed at ground level, within a slightly larger yellow container that was secured to the ground with a peg (Fig. S1). The content of the wind and ground traps were collected weekly. In addition, an iMapPESTS Sentinel model 4 smart suction trap, with a 2 m suction tower (described in *Martoni et al. (2023)*) was deployed at site 1 and collected six daily samples per week. In total, across all sites, 26 trap samples were collected weekly, for a total of 286 expected samples across all traps throughout the duration of the 11-week field trial. All traps collected insects into a 50% solution of Propylene Glycol (PG) and tap water, which has been found to preserve insect samples for non-destructive DNA extraction at a range of temperatures (*Martoni et al., 2021*). The PG solution, being non-flammable, also facilitated shipping of samples from the field to the laboratory.

### Laboratory steps

Samples were collected from the devices weekly by an operator. Environmental data was recorded daily by the iMapPESTS Sentinel smart trap *in loco*, providing precise measurements at the exact geographical location where the insects were collected. Samples were shipped *via* courier to the Agriculture Victoria Research AgriBio facility for metabarcoding analysis. Upon arrival at AgriBio, each weekly batch (26 traps) was immediately processed for DNA extraction and PCR amplification (Fig. 2) following a

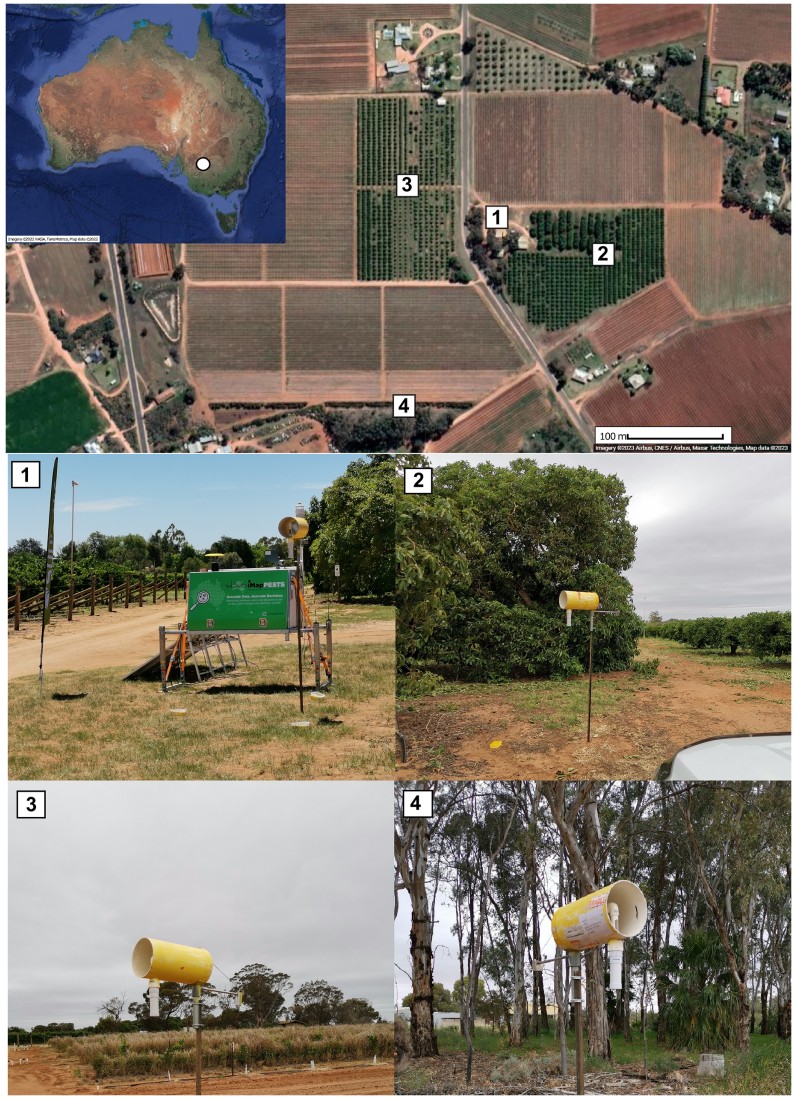

**Figure 1  Map of trial location at the SuniTAFE Smart Farm (Meringa, Victoria, Australia).** Each site had a wind trap and four ground traps, while site 1 also had an iMapPESTS Sentinel smart trap. Site 1 was the car park of the farm (1), site 2 was between a citrus and an avocado orchard (2), site 3 was surrounded by citrus (3), and site 4 was on the margin of a native bush strip (4). Maps were generated using Google Earth v 7.3.6.9345 (Google, CA, USA).           

previously described workflow (*Martoni et al., 2023*). Briefly, samples were filtered to separate the insects from the collection fluid and debris (Figs. 2A–2C) using a 0.2 mm voile polyester fabric mesh that was sterilized before use with a bleach solution (10%) and rinsed with high grade ethanol (100%). While on the mesh, larger insects were separated from smaller specimens and processed as a separate sample in order to partially account for differences in biomass that could cause a bias in the number of metabarcoding reads (*McLaren, Willis & Callahan, 2019*; *Martoni et al., 2022*) (Fig. 2C).

  Non-destructive DNA extraction was performed using the DNeasy Blood and Tissue kit (Qiagen, Hilden, Germany) with an overnight incubation (~17 h) at 56 °C following the

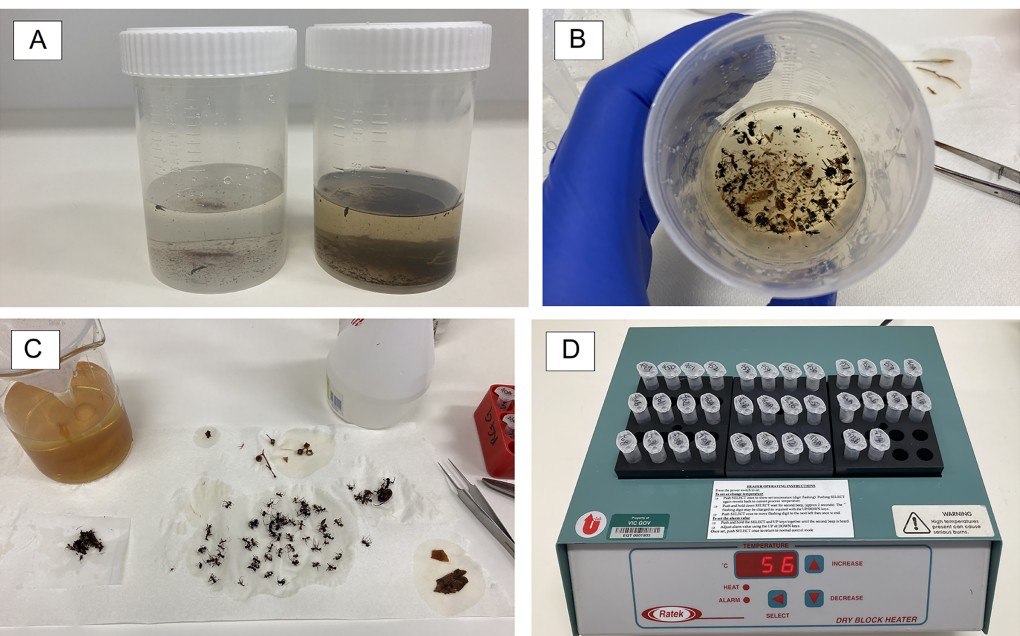

**Figure 2 Sample processing using non-destructive DNA extractions.** Samples conditions varied depending on the different collection traps, with the ground traps being dirtier, often including dust and pieces of plants (A and B, wind trap on the left and ground trap on the right). Glycol was filtered from the sample, debris removed, and larger-sized insects were separated from smaller-sized ones (C). Filtered and sorted samples were then incubated overnight at 56 °C for non-destructive DNA extraction (D).

protocol presented in *Martoni et al. (2021)*. Following DNA extraction, the insect specimens from each trap sample were preserved in high grade ethanol (100%) in case further morphological examination was required. Polymerase chain reactions (PCRs) were performed in duplicate targeting a ~200 bp fragment of the standard barcode region (*Hebert et al., 2003*) within the *Cytochrome c Oxidase* subunit I gene (COI) using the FwhF2-FwhR2n primer pair (*Vamos, Elbrecht & Leese, 2017*). Partial Illumina adapter sequences (underlined) were incorporated into the primers in order to use a 2-step PCR library preparation (FwhF2: 5′ ACACTCTTTCCCTACACGACGCTCTTCCGATC TGGDACWGGWTGAACWGTWTAYCCHCC-3′; FwhR2n: 5′-GTGACTGGAGTTCA GACGTGTGCTCTTCCGATCTGTRATWGCHCCDGCTARWACWGG-3′). The initial PCR was conducted in 25 μL reactions consisting of 14.7 μL of 100 μg/ml Bovine Serum Albumin (New England Biolabs, Ipswich, MA, USA), 5 μL of 5 × Bioline MyFi reaction buffer (Meridian Bioscience, Cincinnati, OH, USA), 1 μL of each primer (10 μM), 0.8 μL MyFi DNA polymerase (Meridian Bioscience, Cincinnati, OH, USA), and 2.5 μL of template DNA. The PCR started with 2 min at 94 °C, followed by 30 cycles of denaturation at 94 °C for 30 s, annealing at 49 °C for 45 s and extension at 72 °C for 45 s. Finally, the reaction ended with an additional extension phase of 2 min at 72 °C. Amplification was verified by gel electrophoresis (1% w/v agarose). Amplified libraries were stored in a −20 °C freezer until a full 96 well microtiter plate (47 trap samples in duplicates +1 DNA
extraction and 1 PCR controls = 96 samples) could be simultaneously processed for real time PCR (rtPCR), normalization and sequencing.

Library preparation followed a previously described workflow (*Martoni et al., 2023*). Specifically, after the first PCR, amplicons were diluted 1/10 before a second round of rtPCR was used to attach 8 bp unique dual indexes and the remainder of the Illumina adapter sequences. Each indexing rtPCR reaction (50 μL volume) contained 32.5 μL of ddH$_2$O, 10 μL of 5 × Phusion HF Buffer (Thermo Fisher Scientific, Waltham, MA, USA), 1 μL of dNTP mix (10 mM), 1 μL of SYBR Green I Mix (Thermo Fisher Scientific, Waltham, MA, USA) diluted 1/1,000 in ddH$_2$O, 0.5 μL Phusion DNA polymerase (Thermo Fisher Scientific, Waltham, MA, USA), 4 μL of sample-specific indexes (2.5 μM), and 1 μL of the diluted amplicon from the first PCR. The cycling conditions for rtPCR were 98 °C for 30 s, followed by 6–8 cycles of 98 °C for 10 s, 65 °C for 30 s, and 72 °C for 30 s, with fluorescence measurement conducted in the 65 °C and 72 °C phases. Amplification curves were monitored in real time and the reactions stopped while still in the exponential phase to prevent over-amplification artefacts.

Library checks and sequencing also followed the workflow presented in *Martoni et al. (2023)*. After rtPCR, the libraries were purified and normalized using the SequalPrep™ Normalization Plate Kit (Thermo Fisher Scientific, Waltham, MA, USA) following the manufacturer's protocol, but eluting the final product in 15 μL instead of 20 μL. Normalised and cleaned libraries were then pooled together and the resulting pool was quality checked and quantified using a High Sensitivity D1000 ScreenTape assay performed on a 2200 TapeStation (Agilent Technologies, Santa Clara, CA, USA). The final pooled library was diluted to a concentration of 7 pM, spiked with 15% PhiX and sequenced using V3 chemistry (2 × 250 bp reads) on an Illumina MiSeq system (Illumina, CA, USA).

## Bioinformatic analysis

Bioinformatic analysis followed the pipeline available at: https://alexpiper.github.io/iMapPESTS/local_metabarcoding.html, as reported in *Martoni et al. (2023)*. Raw sequence reads were demultiplexed using bcl2fastq allowing for a single mismatch in the indexes, then trimmed of PCR primer sequences using BBDuK v38 (*Bushnell, Rood & Singer, 2017*). Sequence quality profiles were used to remove reads with more than one expected error (*Edgar & Flyvbjerg, 2015*), or those containing ambiguous 'N' bases, with all remaining sequences truncated to the length of the target amplicon (205 bp) and analysed using DADA2 v1.16 (*Callahan et al., 2016*). Following denoising, amplicon sequence variants (ASVs; *Callahan, McMurdie & Holmes, 2017*) inferred separately from each sequencing run were combined into a single table and chimeras were detected and removed de-novo using the *removeBimeraDenovo* function in DADA2. To further filter any non-specific amplification products and pseudogenes the ASVs were aligned to a profile hidden Markov model (PHMM) (*Eddy, 1998*) of the full-length COI barcode region from (*Piper et al., 2021*), and then checked for frame shifts and stop codons that commonly indicate pseudogenes (*Roe & Sperling, 2007*). Taxonomy was assigned using the IDTAXA algorithm of *Murali, Bhargava & Wright (2018)* implemented in the DECIPHER v2.22.0 R

package, which was trained on an in-house insect and arachnid COI database (*Piper et al., 2021*), accepting only assignments with a bootstrap confidence threshold of 60% or above. To increase classification to species level, we also incorporated a BLAST v2.13.0 (*Altschul et al., 1990*) search and, to reduce the risk of over-classification, only accepted BLAST species assignments if the BLAST search agreed with IDTAXA at the Genus rank. Finally, all retained ASVs assigned to the same insect species were merged, while ASVs that could only be assigned to a higher taxonomic rank (*i.e.*, genus, family, order) were manually blasted on GenBank (*Sayers et al., 2022*). This enabled us to match, or partially match, the ASVs to sequences present on GenBank that may have been uploaded more recently than the reference database was created, or did not pass the stringent filtering parameters defined in *Piper et al. (2021)*. ASVs matching GenBank sequences with a similarity between 99–100% were labelled using the accession number of the respective match (*e.g.*, Diptera sp. XX00000). ASVs partially matching sequences, with a similarity between 96–98.99%, were labelled as "near" the accession number of the GenBank sequence they matched (*e.g.*, Diptera sp. nr XX00000). Following this procedure, ASVs with a genetic similarity <96% to any given sequence present on GenBank were manually aligned using Geneious Prime® 2022.0.2 (www.geneious.com) and MEGA X (*Kumar et al., 2018*) and grouped into a single operational taxonomic unit (OTU) when their divergence was <5%. Samples with less than 2,000 reads, as well as ASVs with <0.01% relative abundance in each sample, were discarded from the dataset.

The R v4.1.0 statistical computing environment (*R Core Team, 2023*) was used to perform statistical analysis using the *ALDEx2* v1.26.0 (*Fernandes et al., 2013*), *phyloseq* v1.38.0 (*McMurdie & Holmes, 2014*), *tidyverse* v1.3.1 (*Wickham et al., 2019*) packages, as well as to generate Figs. 3–7. Species accumulation curves were generated using the *vegan* v2.6-4 R package (*Oksanen et al., 2020*) to confirm appropriate sequencing depth was achieved to characterise the diversity within each sample, and the observed number of ASVs was also compared to richness estimates obtained using *breakaway* v4.7.9, which estimates the number of species may have been missed due to insufficient sampling (*Willis, Bunge & Whitman, 2017*). Heat trees were generated using the *Metacoder* v0.3.5.1 package (*Foster, Sharpton & Grünwald, 2017*), both as a graphic representation of the taxonomic diversity (Fig. 3), and for the comparison of traps and sites (Fig. 7). Additional details of Fig. 7 were created using BioRender (BioRender.com). Venn diagrams showing species overlap between the different traps and sites (Fig. 5) were generated using the *MicEco* v0.9.17 R package (*Russell, 2021*). Finally, Figs. 4 and 6 were generated using the *ggplot2* v3.3.5 R package (*Wickham, 2016*).

## Statistical analysis

For α-diversity measures, we used three complementary metrics to account for phylogenetic distance (phylogenetic diversity–pd; *Faith, 1992*) and abundance (Shannon; *Shannon, 1948*), as well as simple presence-absence (Observed). Similarly, for β-diversity analysis we used four distance metrics (Jaccard, Bray-Curtis, Aitchison, and Philr) in order to consider not only presence-absence of taxa (Jaccard index; *Jaccard, 1912*), but also their read counts (Bray-Curtis index; *Bray & Curtis, 1957*), relative abundance within a

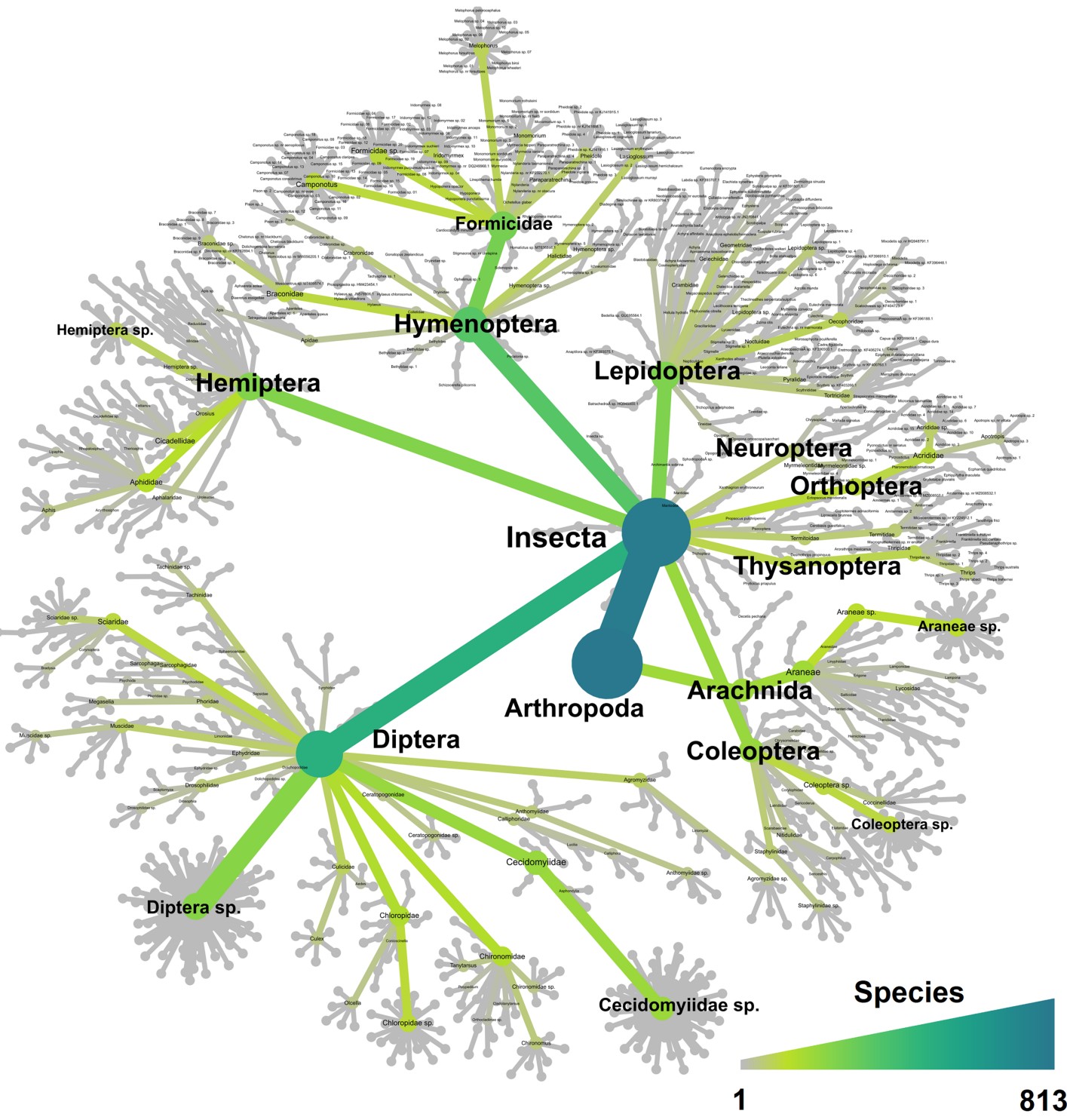

**Figure 3 Heat tree showing all the 813 taxa recorded in this study and their taxonomical distribution.** The tree is colour-coded based on the number of species recorded for each taxon.

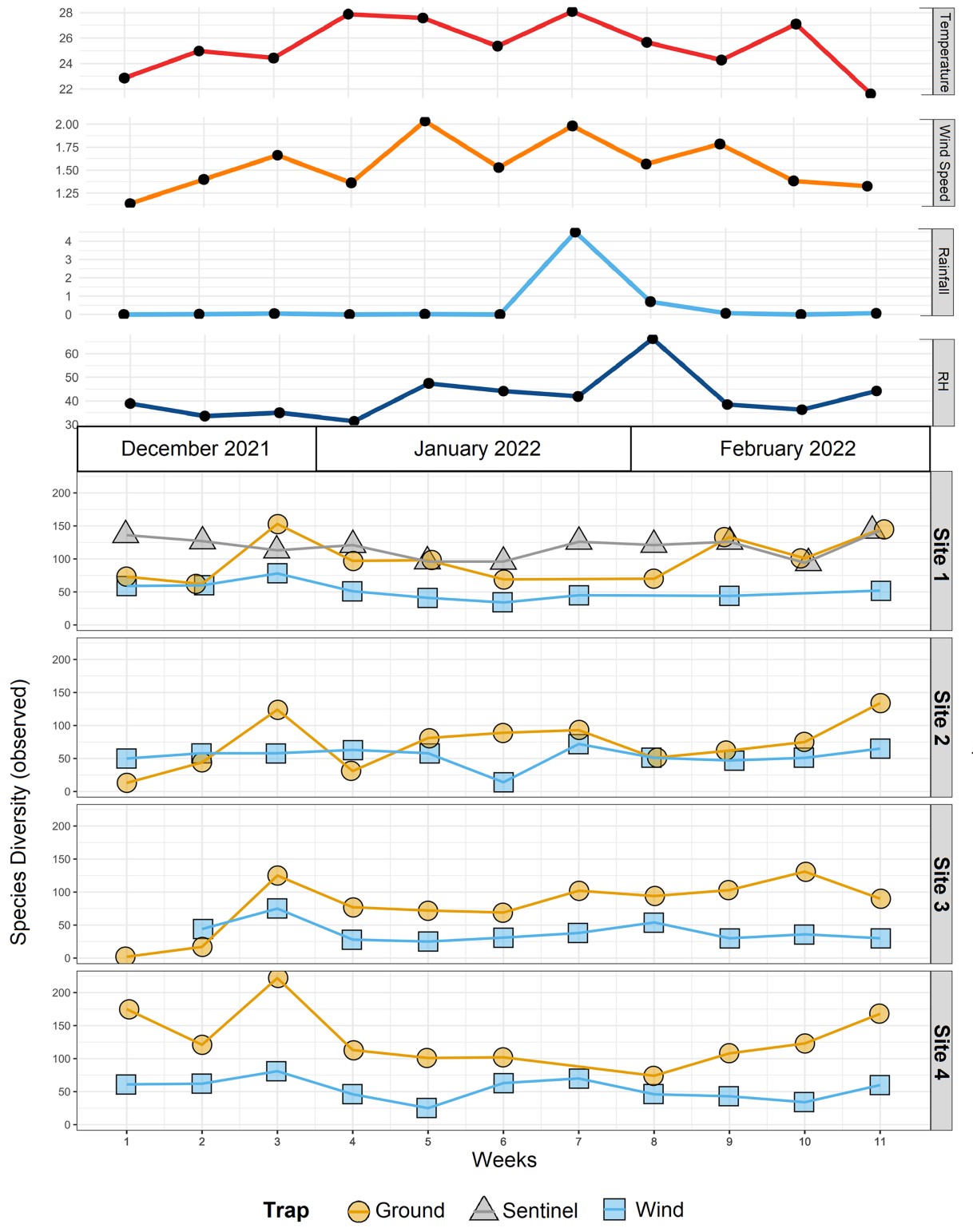

**Figure 4 Species diversity across sites and traps, and environmental data.** Environmental data (mean temperature, wind speed, rainfall, and relative humidity-RH) associated with observed α-diversity across weeks, by different traps (shapes) and sites.

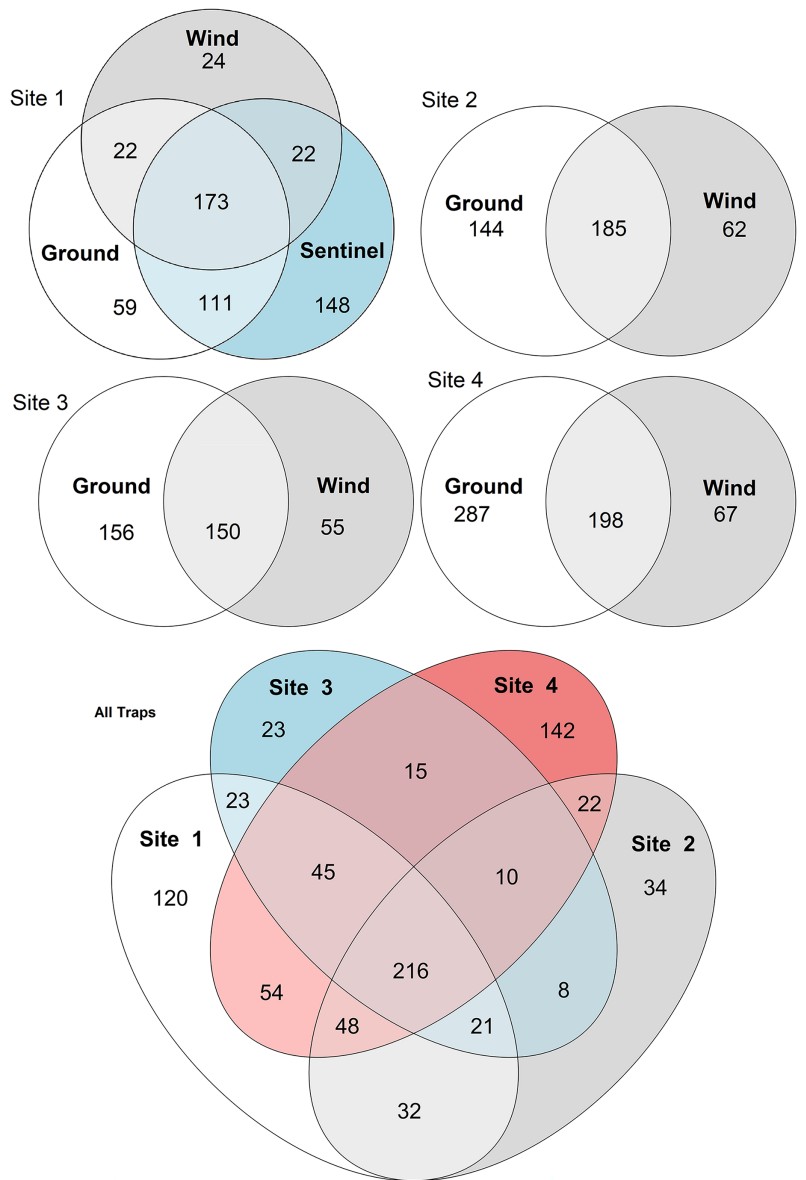

**Figure 5 Venn diagrams of species diversity across traps and sites.** The diagrams show the number of species collected by each trap type across sites (sites 1–4, above), and the number of taxa collected across sites by all traps (below). Numbers represent individual taxa.

compositional data analysis framework (Aitchison index; *Aitchison et al., 2000*), and phylogenetic divergence as well as relative abundance within a similar compositional framework (Philr; *Silverman et al., 2017*). Principal coordinates analysis (PCoA) was used to graphically represent relationships between samples in multidimensional space using the β-diversity dissimilarity matrices. Finally, we compared β-diversity between the trap types using permutational multivariate analysis of variance (PERMANOVA) tests with the adonis2 function from the *vegan* R package (*Oksanen et al., 2020*), with one test comparing

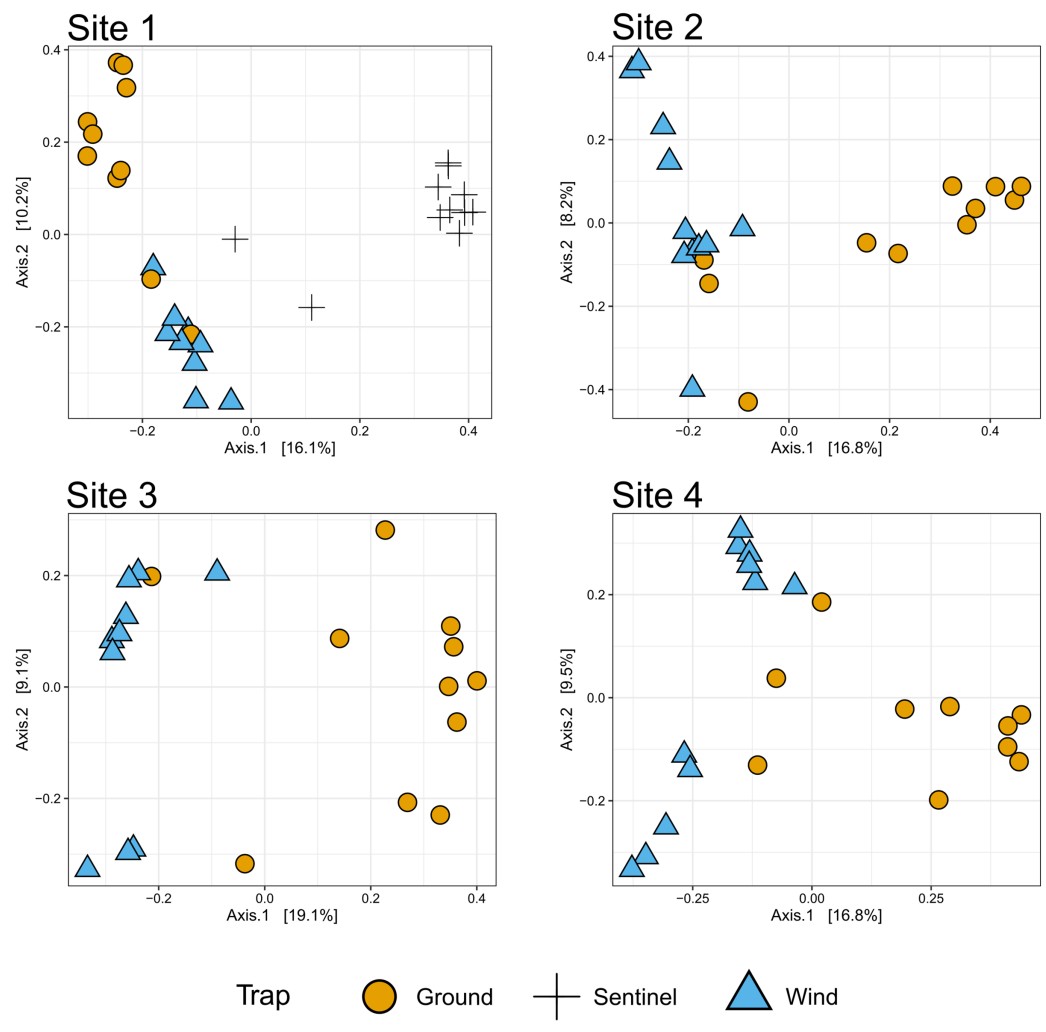

**Figure 6 β-diversity across sites.** The PCoA plots (Bray-Curtis distance) show the insect diversity collected by each trap for each site of the trial.

all trap types from site 1 only (*i.e.*, including the sentinel), and a second test comparing just the wind and ground traps across all sites.

## Insect morphological identification

Morphological examination of psyllid (Hemiptera: Psylloidea) specimens was conducted post non-destructive DNA extraction to confirm the presence of a *Casuarinicola* species, which was reported by the metabarcoding analysis within some samples (see below). Morphological identification was based on the taxonomical keys provided by *Taylor et al. (2010)*. Images of adult psyllids and of the exuviae of an immature were obtained stacking together from five to 30 images obtained using a Leica stereo microscope M205C with a DFC450 camera using the Leica Application Suite software v4.5.0. These images were used to generate Fig. 8.

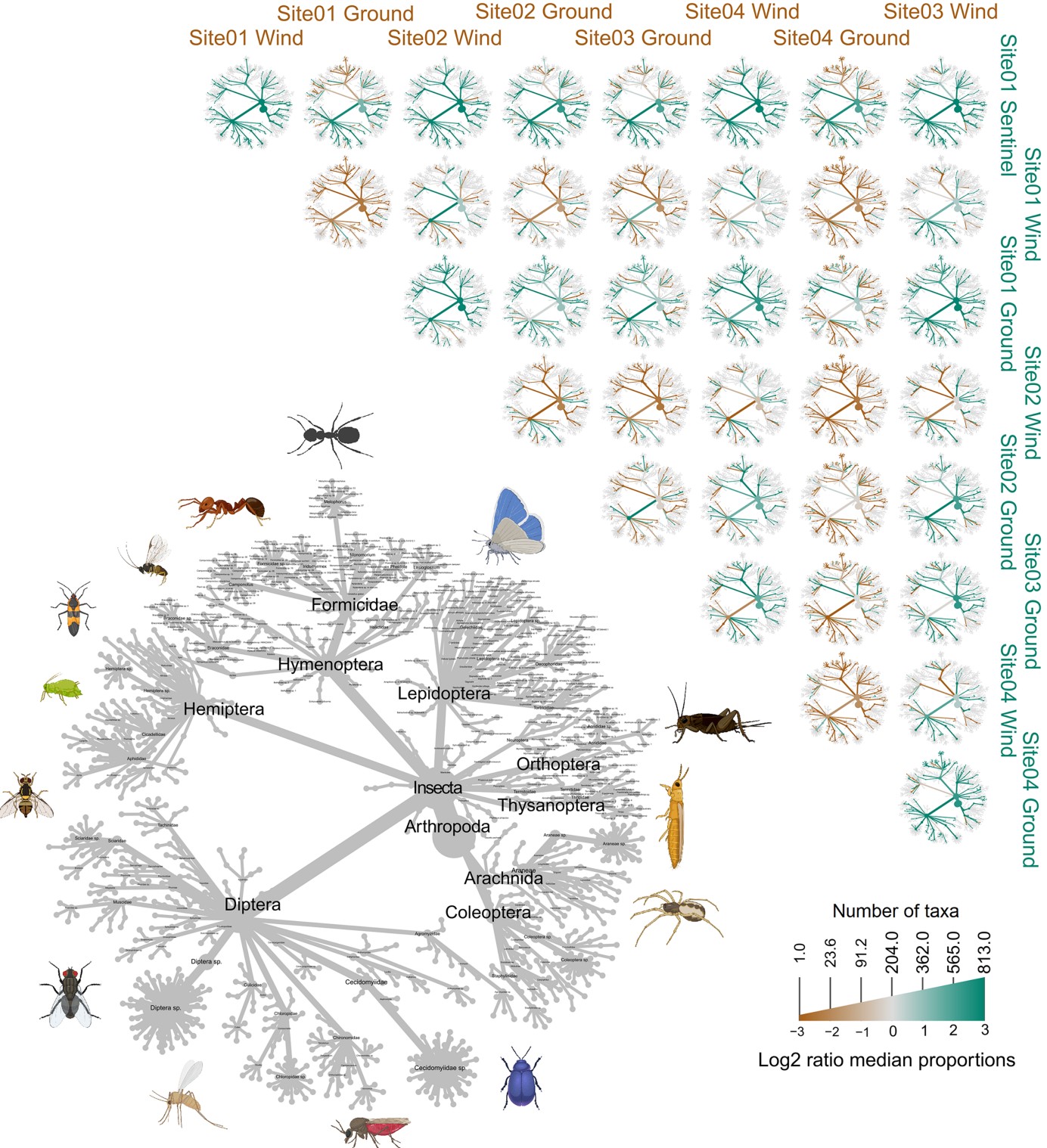

**Figure 7 Heat trees comparing the insect diversity recorded by each sample, subdivided by site and trap type.** The grey tree on the lower left functions as a key for the unlabeled trees. Each of the smaller trees represent a comparison between two samples, each from a specific site and trap. A taxon branch colored brown is more abundant in the sample with the brown label, and a taxon branch colored green is more abundant in the sample with the green label. Parts of this figure were created with BioRender.com.

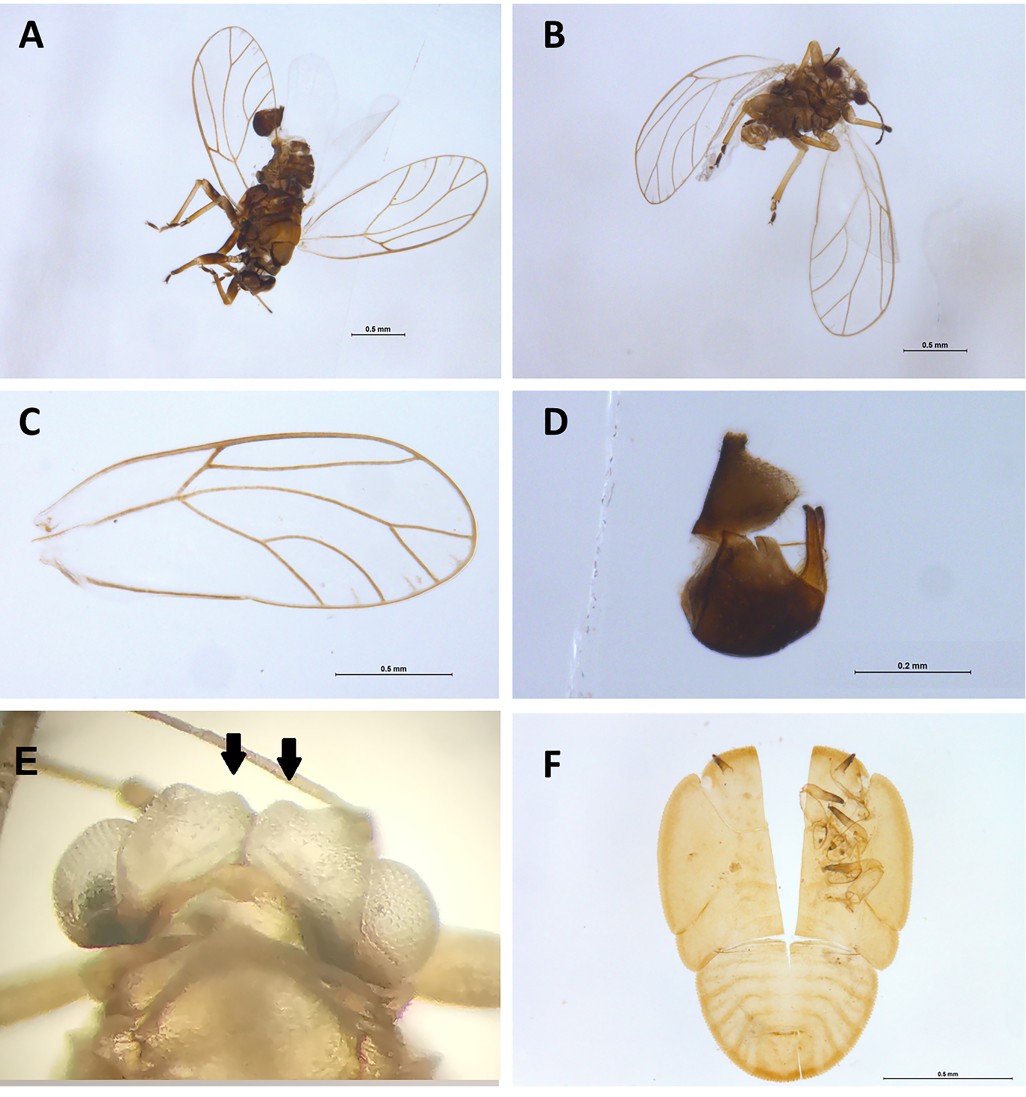

**Figure 8 Specimens of *Casuarinicola* cf *warrigalensis* recovered in this study.** Two adults (A–E) and the exuviae of a nymph (F) were recovered. Morphological examination of the insect characters such as wing (C), male genitalia (D), head (E) and genal processes (indicated by arrows) following non-destructive DNA extraction enabled confirmation of insects' identity. Scale bars are 0.5 mm (A–C and F) or 0.2 mm (D).

# RESULTS

## Metabarcoding results

The trial was expected to generate a total of 286 samples across the 11 weeks. However, due to environmental events, including a storm with strong winds on week 1 (traps were windblown), heavy rainfall on week 7 (trap content overflown, Fig. 4), and high temperatures recorded on some days (trap preservative evaporated), only 271 samples were received for identification. Of these, 66 were collected by the Sentinel, 44 from the wind traps, and 161 from the ground traps. The missing 15 samples were all from the

ground traps, which were strongly impacted by the weather conditions. After quality control, 54,314,023 sequence reads were retained from 65 Sentinel samples, 40 wind trap samples and 149 ground trap samples. These were assigned to a total of 813 taxa, from at least 354 genera, 148 families and 21 orders, across insects and arachnids (Fig. 1 and Table S1). The insect order with the most diversity was Diptera with 314 taxa (38.62%), followed by Hymenoptera (158 taxa, 19.43%), Lepidoptera (88 taxa, 10.82%), Hemiptera (82 taxa, 10.09%), and Coleoptera (56 taxa, 6.89%) (Fig. 3). Of the 813 detected taxa, 280 (34.44%) matched and 20 (2.46%) nearly matched (>96% similarity) the COI sequence of a described species, with an additional 46 taxa (5.66%) and 32 taxa (3.94%) respectively matching or nearly matching a COI sequence available on GenBank that was not identified to species level (*e.g.*, Diptera sp. XX00000 or Diptera sp. nr XX00000) (Fig. 3, "Diptera sp."). Of the remaining 435 taxa, 118 taxa (14.51%) could only be assigned to a genus, 192 (23.62%) only to a family, 123 (15.13%) only to an order, while two taxa (0.25%) could only be assigned to a class.

## Comparison of diversity between sites and trap types

To compare the insect diversity and community composition collected by all three trap types, only site 1 was considered since this was the only site to have all three traps present. For this comparison, the daily Sentinel samples were merged by week to standardize the collection time with the ground and wind trap samples which were only collected weekly, as is custom for these traps (*Martoni et al., 2023*). Across all weeks, the Sentinel collected the highest α-diversity within site 1, recording 454 species over the course of the study, while the ground trap collected 365 species and the wind trap only 241 (Fig. 5). There were, however, differences seen between weeks, with the Sentinel collecting more species for 8 of the 11 weeks, while the ground trap collected the highest diversity for the remaining three weeks (Weeks 3, 9 and 10; Fig. 4). Comparisons of species composition (β-diversity) between the three trap types at site 1 revealed significant differences in presence absence of species (Jaccard), and presence/absence plus abundance (Aitchison and Bray-Curtis), between traps, with between 11.9–22.7% of the variance in sample composition explained by the trap type (Jaccard: $R^2 = 0.119$, $p = 0.001$; Bray-Curtis: $R^2 = 0.227$, $p = 0.001$; Aitchison: $R^2 = 0.184$, $p = 0.001$). However PERMDISP tests also found significant differences in dispersion between the groups, especially when presence/absence plus abundance were taken into account (Jaccard: $F_{(2,27)} = 6.013$, $p = 0.005$; Bray-Curtis: $F_{(2,27)} = 8$, $p = 0.001$; Aitchison = $F_{(2,27)} = 39.75$, $p = 0.001$), suggesting that differences in community composition was also influenced by differences in composition within groups (Fig. S2–S4). This difference in within-group dispersion can be seen on the PCoA plot for site 1 (Fig. 6), where the wind traps are more tightly clustered (low dispersion) while the Sentinel and Ground traps had some samples that were separated from the main clusters (high dispersion). These outlier samples for Ground and Sentinel traps appear to be linked to the first 2–3 weeks of the trial, during which these traps tended to collect lower diversity, making the trap catch more similar of that of the wind trap (Fig. 4). This is probably due to the lower number of insects active early in the season, when temperatures were cooler and wind speeds lower. When taking into account the phylogenetic relatedness between species
along with their relative abundances (Philr), the communities caught by each trap were less similar ($R^2 = 0.267$, $p = 0.001$), and PERMDISP found no significant differences in within-group dispersion between the three traps ($F_{(2,27)} = 0.71$, $p = 0.486$) (Figs. S2–S4).

When considering all sites, but just the wind and ground traps, the insect diversity collected from the ground trap was generally higher than the wind trap across the course of the 11 weeks, with the ground trap collecting 329 distinct species compared to 247 by the wind trap at site 2, 306 species to 205 species at site 3 and 485 species to 265 at site 4 (Figs. 4 and 5). The only exceptions to this trend were at site 2 in weeks 1, 2 and 4; and at site 3 in week 2, where the wind trap collected more diversity than the ground trap (Fig. 4). A small (4.2–9%) but significant difference in community composition (β-diversity) was recorded between different traps (Jaccard: $R^2 = 0.042$, $p = 0.001$; Philr: $R^2 = 0.090$, $p = 0.001$), with a clear separation seen in the PCoA plots generated, especially when comparing the different traps across sites (Fig. 6).

When comparing the arthropod diversity between sites, the Sentinel trap was excluded from the dataset as it was only present at site 1. The combined catch of wind and ground traps across all sites showed a smaller (5.2–7.2%) but significant difference in β-diversity (Jaccard: $R^2 = 0.052$, p = 0.003; Bray-Curtis: $R^2 = 0.072$, $p = 0.001$). When considering only species collected by the wind and ground traps, site 4, the native bush, recorded the highest diversity across all weeks (552 species) with a different taxonomic composition compared to other sites (Fig. 7). This site also showed the highest number of species that were not recorded in other sites (142 species, Fig. 5). Site 1, the car park, recorded the second highest diversity (411 species), followed by site 2, the semi-grassy area between a citrus orchard and avocados (391 species), and site 3, the orchard containing only citrus (361 species) (Fig. 5). Only 216 taxa were detected from all sites during the 11-week trial, even when including the Sentinel samples (Fig. 5). When disentangling the insect diversity collected by the different traps at different sites based on the insect taxonomy, the ground traps recorded taxa from the families Formicidae and Acrididae (Fig. 7), that were generally not collected by other traps.

### Beneficials, pests and target species

The main target of this study was the superfamily Psylloidea, with the aim to verify the absence of high-priority exotic pests, as well as assess the diversity of Australian psylloids around citrus orchards. Within the superfamily Psylloidea, seven species were recorded across all three trap types: *Anoeconeossa communis*, *Blastopsylla occidentalis*, *Cryptoneossa triangula*, *Ctenarytaina eucalypti* and *Eucalyptolyma maideni* (all Aphalaridae), *Acizzia acaciaebaileyanae* (Psyllidae) and a *Casuarinicola* species (Triozidae). The initial molecular-based identification obtained for the *Casuarinicola* species using the COI data publicly available on GenBank was based on a match (97.07% genetic similarity) with the exotic species *Casuarinicola novacaledonica* Taylor, known to be present only in New Caledonia (*Taylor et al., 2010*). DNA reads (14,129) of this species were detected across four samples, with 2,866 reads, 11,122 reads (6,849 and 4,273 in the two replicates, respectively), 119 reads and 22 reads per sample. When revisiting the non-destructively extracted specimens, individual adult *Casuarinicola* (Figs. 8A–8D) were found in the two

samples which recorded the highest number of reads for this taxon, while the exuviae of an immature psyllid was found in the sample with 119 reads (Figs. 8E–8F). Follow-up morphological examination of these specimens, following the existing keys to the species of *Casuarinicola* (*Taylor et al., 2010*), suggested the species in question did not match *C. novacaledonica*, due to the shape and size of the wing, the length of the genal processes (Fig. 8, arrows) and the shape of the male terminalia. The morphological characters (Fig. 8), well-preserved even after non-destructive DNA extraction, enabled the identification of this species as *Casuarinicola* cf *warrigalensis*, an Australian endemic species that had not been reported in Victoria prior to this study. No specimens of this triozid could be found in the sample which had only 22 reads for this taxon, however a predatory Hemiptera sp. (morphologically identified as a *Microvelia* sp., Veliidae) was present in this sample and could potentially be the cause of the reads through predation on this species. Nevertheless, this low number of reads could also arise through cross-sample contamination due to index switching, despite the application of a 0.01% minimum abundance threshold to control for this (*Piper et al., 2022*). No DNA sequences of exotic psylloid pests of citrus, including the Asian citrus psyllid, *Diaphorina citri*, or the African citrus psyllid, *Trioza erytreae*, were recorded here.

While no psylloid pests were recorded during this trial, pests from various other insect groups that are known to damage citrus were identified across all trap types (Fig. 3). In particular, the Citrus leaf miner, *Phyllocnistis citrella* Stainton, Kelly's Citrus thrips, *Pezothrips kellyanus* (Bagnall), the plague thrips, *Thrips imaginis* Bagnall, the spiraea aphid, *Aphis spiraecola* (Patch), and the Queensland Fruit Fly, *Bactrocera tryoni* (Froggatt) were all detected during the 11-week field trial. In addition to the insect pests, the two-spotted mite *Tetranychus urticae* C.L.Koch, a pest of citrus, was also reported in the trap samples. Other pests recorded here but not known to be associated with citrus encompassed a number of insect orders (Table S1). Amongst thrips (Thysanoptera), we recorded the Western Flower thrips, *Frankliniella occidentalis* Pergande, and the flower thrips *Frankliniella schultzei* (Trybom). Within Hemiptera, more than 30 aphid species were detected, including the lettuce aphid *Acyrthosiphon lactucae* (Passerini), the coriander aphid *Hyadaphis coriandri* (Das), the wild crucifer aphid *Lipaphis erysimi* (Kaltenbach), the green peach aphid, *Myzus persicae* (Sulzer); but also the greenhouse whitefly, *Trialeurodes vaporariorum* Westwood. Amongst beetles (Coleoptera), we recorded the 28-spotted potato ladybird, *Henosepilachna vigintioctopunctata* (Fabricius), the cucurbit ladybird, *Henosepilachna sumbana* (Bielawski), the dried-fruit beetle, *Carpophilus hemipterus* (Linnaeus), and the pineapple beetle, *Urophorus humeralis* (Fabricius).

Amongst the beneficial insects recorded, we reported a number of predatory ladybird beetles (Coccinellidae), including *Coccinella transversalis* Fabricius, *Hippodamia variegata* (Goeze) and *Diomus notescens* (Blackburn). We recorded syrphid hoverflies (Diptera), the larvae of which are known predators of insect pests such as aphids: *Simosyrphus grandicornis* (Macquart) and *Sphaerophoria macrogaster* Thomson. Amongst the more than 20 taxa of the parasitoid family Braconidae, we recorded the parasitoid wasp *Dolichogenidea tasmanica*, as well as the light brown apple moth, *Epiphyas postvittana*

(Tortricidae) this wasp is used as a biological control agent for. We also recorded the parasitoid wasps *Diadegma rapi* (Ichneumonidae) and *Apanteles ippeus* (Braconidae), known to parasitise the diamond back moth, *Plutella xylostella* (Plutellidae), which we also recorded at high numbers. Finally, we recorded the lacewing (Neuroptera) predators *Mallada signatus* and *Micromus tasmaniae*, and the crusader bug *Mictis profana* (Coreidae), a hemipteran biocontrol agent.

## DISCUSSION

### The benefits of non-destructive DNA extraction for biosecurity

For this trial, the main insect targets were members of the superfamily Psylloidea, as surveillance for high-priority exotic psylloid pests of citrus is a critical task for the Australian citrus industry. While no native Australian psylloid is known to be a pest of citrus (*Hollis, 2004*), psylloids can be easily windblown onto plants other than their hosts, often in high numbers, raising false alarms for growers (*Martoni & Blacket, 2021*). In this study we confirmed that exotic citrus pests such as the African and the Asian citrus psyllids were not present within the Sunraysia region. However, we did record seven Australian native psylloid species, collected across all trap types, from seven genera: *Acizzia* (Psyllidae), *Anoeconeossa*, *Blastopsylla*, *Cryptoneossa*, *Ctenarytaina*, *Eucalyptolyma* (all Aphalaridae) and *Casuarinicola* (Triozidae). These psyllids are mostly associated with *Eucalyptus* spp. (Aphalaridae) but also with *Acacia* spp. (Psyllidae) and *Casuarina/Allocasuarina* spp. (Triozidae), confirming there is potential for these insects to be windblown into citrus crops from nearby host plants, which were present at site 4. This information on psyllids dispersing into citrus crops could prove useful for future studies aiming to determine whether Australian native psylloids can vector and/or transmit plant pathogens normally harbored by exotic pest species. Indeed, in recent years, Australian native species of the genus *Acizzia* (*Morris et al., 2017*) and New Zealand species of *Ctenarytaina* (*Thompson et al., 2017*) have been found to vector non-pathogenic strains of *Candidatus* Liberibacter, raising the question whether native psylloids could potentially become vectors of the plant pathogenic bacteria vectored by exotic pests.

This study further demonstrated that insect metabarcoding can be an extremely powerful tool for revealing the geographic distribution of incidentally detected species that were not the main target of the surveillance efforts. Here we recorded, for the first time in Victoria, the non-pest psyllid *Casuarinicola* cf *warrigalensis*, previously reported only from New South Wales (*Taylor et al., 2010*). The identification of this species was made possible by the use of a non-destructive DNA extraction method, which allowed us to retain intact specimens with well-preserved morphological characters. The presence of this species in Victoria is consistent with the distribution of its host plant, *Casuarina cristata*, across the inland eastern Australian region (G. Taylor, 2023, personal communication). The record of this triozid species, which was not the target of the surveillance efforts for the citrus industry, demonstrated that metabarcoding can detect the presence of a previously unreported insect, even at an immature stage, through the whole-community information provided by this technique. The utility of the non-destructive metabarcoding approach has previously been demonstrated in a similar way for aphid surveillance, identifying a new
record for Australia based on the metabarcoding detection and morphological confirmation of a single nymph (*Batovska et al., 2021*). However, the detection of *C.* cf. *warrigalensis* also highlights the importance of expanding and curating COI databases for poorly studied insect groups. With a genetic similarity of >97% to the only available sequence of *Casuarinicola novacaledonica*, an exotic species, the *C.* cf. *warrigalensis* specimens recorded here were initially flagged as a potential exotic detection. This example highlights the importance of continued generation of high-quality barcode sequences for poorly studied insect groups in order to increase the effectiveness of molecular identification (*Piper et al., 2019*). The Australasian Psylloidea are known to be under-represented in public nucleotide databases, with recent barcoding studies highlighting a higher-than-expected species diversity (*Martoni et al., 2018*; *Martoni, Taylor & Blacket, 2020*). These represent just one of the many insect groups from under-sampled regions such as Australia that require attention in future taxonomic and DNA barcoding efforts (*Taxonomy Decadal Plan Working Group, 2018*).

## Insect diversity across sites and traps

Insect collection methods have been studied for decades, with relative effects of sampling methods as well as habitat types being very well-known (*Juillet, 1963*; *Disney et al., 1982*; *Duelli, Obrist & Schmatz, 1999*; *Missa et al., 2009*; *Aguiar & Santos, 2010*; *Russo et al., 2011*). The metabarcoding analysis showed how traps with different collecting mechanisms preferentially detect different insect taxonomic groups, reinforcing the results of previous morphological identification based studies (*Noyes, 1989*; *Agosti et al., 2000*; *Basset et al., 2003*; *Lamarre et al., 2012*; *Matos-Maraví et al., 2019*). In particular, the insects collected using air sampling (*i.e.*, Sentinel and wind) were substantially different to those collected within ground traps. While most of the insects collected only from the ground traps were ants and crickets, not of concern to the citrus industry, this served as a reminder that there is no single trap capable of comprehensively assessing diversity across the insect tree of life. Additionally, different trapping methods can affect the sample processing of the metabarcoding workflow, too. The ground traps, being completely exposed to the environment, were greatly affected by environmental conditions, including strong winds, rainfall and, in one instance, also being run over by a tractor. This not only meant that 15 out of 176 ground trap samples (8.52%) could not be collected from the field, but it also meant that a further 12 samples (6.81%) could not be processed due to a high level of debris (*e.g.*, plant material, dust, soil; Fig. 2), possibly working as a DNA extraction and PCR inhibitor. The 15.34% of ground trap samples failing library preparation is a striking contrast to the 9% of wind trap samples that failed and, only 1.51% of Sentinel trap samples, with the latter generally related to low number of insect specimens present in the sample. Sample condition is extremely important for insect metabarcoding analysis, especially when applied to high throughput biosecurity surveillance, where commonly a high volume of samples need to be rapidly processed. In this context, having to repeat DNA extraction or PCR delays the generation of time-sensitive data and impedes any response to pest detections. On the other hand, the results presented here show how many ground-dwelling insects are not observed in wind-based traps, making the use of multiple

complementary trapping approaches, including ground traps, fundamental for assessing the general insect biodiversity of an area.

Differences in insect diversity and species composition were also found between different sites from the same farm, indicating that sampling from a single site is insufficient for characterizing the broader species composition within an area, and may miss pests that are present in low abundance. Indeed, despite being only a few hundred meters apart, different areas of the same farm may host different plants and, therefore, different species of insects associated with them. Ground traps placed near the strip of native vegetation at site 4 collected the highest diversity of insects across the 11 weeks, while on the other hand similar composition of the wind traps could be observed across all sites. This can probably be explained by the fact that insect diversity collected by passive wind traps almost entirely depends on the wind. Since this trap is not actively aspirating insects, most of the strong flyers can probably avoid being captured (*Teulon & Scott, 2006*), while the insects that are sampled are blown by the wind in similar volumes (and similar diversity) across the whole farm. On the other hand, ground traps appeared to collect a more site-specific insect diversity, suggesting that the distribution of ground dwelling insects is more localized. Both sentinel and ground traps always collected higher diversity compared to the wind traps at site 1 (Fig. 4). This result confirms the improved sampling strength of the Sentinel smart trap over previous trap designs, which was also demonstrated in a previous study comparing it to an alternative suction trap (*Martoni et al., 2023*), but it also highlights the importance of ground traps as a complementary method. From a pest surveillance and biosecurity perspective, the results obtained here highlight the importance of understanding the ecology and biology of the insects targeted, since not only different traps but also different sites within the same farm may produce different results. These findings add to the understanding that spatial replication of traps is critical for assessing the presence/absence of species in an area, since single traps are likely to miss taxa, especially at low population levels (*Berec et al., 2015*; *Montgomery et al., 2021*).

## Non-target insect pests and beneficials

The importance of understanding the interactions between insect pests and beneficials in citrus orchards has been discussed for many decades (*Lo & Chiu, 1988*; *Liang & Huang, 1994*; *Silva et al., 2010*) and across many countries (*Jacas & Urbaneja, 2010*; *Niu et al., 2014*; *Gama, 2017*). Indeed, understanding the arthropod communities present in an orchard can be a very important tool for biological control and integrated pest management approaches, as well as for conservation (*Niu et al., 2014*). As we previously reported (*Martoni et al., 2023*), the combination of non-targeted surveillance traps such as the Sentinel along with non-destructive metabarcoding techniques provides opportunities for revealing insect community composition while at the same time detecting target and unexpected pest species. Indeed, while the main focus of this work were psylloids, we additionally recorded a large variety of non-target pests and beneficials insects and arachnids, across diverse genera, families and orders. Most relevant for this trial were the detections of various non-psyllid species known to be pests of citrus, including the Citrus leaf miner, *Phyllocnistis citrella*, Kelly's Citrus thrips, *Pezothrips kellyanus*, the plague

thrips, *Thrips imaginis*, the spiraea aphid, *Aphis spiraecola*, and the Queensland Fruit Fly, *Bactrocera tryoni*. Even more interesting was the record of a mite pest of citrus, the two-spotted mite, *Tetranychus urticae*, highlighting that the generic insect COI primers used here could successfully amplify and detect arachnids, too. The records of other non-citrus pests, including thrips, aphids, whiteflies, and beetles, suggests that non-targeted metabarcoding of wind and suction traps may provide valuable information on pest presence with relevance beyond the farm and cultivar they were deployed on. With enough sampling sites, metabarcoding datasets could potentially monitor the presence and distribution of pests across a whole growing region to inform area-wide management approaches, independently of the crops that have been targeted.

Similarly, the record of beneficial insects and the general information on insect biodiversity could prove to be extremely useful not only for biosecurity, but also for biological control and ecological studies (*Niu et al., 2014*). Predatory ladybird beetles (Coccinellidae), commonly used as biocontrol agents for a variety of insect pests including psyllids and aphids, were frequently recorded across the 11-week field trial. Similarly, hoverfly larvae detected in the metabarcoding data are very useful predators of aphids, and the crusader bug, *Mictis profana*, which is considered a potential biocontrol agent for the weed *Mimosa pigra* (*Flanagan, 1994*), was also detected.

One of the main limitations reported here remains the lack of DNA sequence availability for poorly studied insect groups, highlighted by the fact that less than 40% of taxa detected in this study could be assigned to species. While our inability to assign many of the metabarcoding sequences to species level and the detection of multiple undescribed or poorly known taxa is concerning, this also presents a potential opportunity for future studies, as detection of a COI barcode alone can provide a foundation for further ecological and taxonomic investigation (*Ratnasingham & Hebert, 2013*). Being able to obtain a DNA sequence for these insufficiently identified taxa can allow them to be linked across studies, localities and time, and eventually barcoded or described. For example, we recorded here a specimen belonging to the genus *Toya* (Hemiptera: Delphacidae) that matched (100%) a sequence generated from a taxon recorded on Barrow Island, one of Western Australia's most important conservation reserves (*Gopurenko et al., 2013*). Another instance was a *Mesocentrus* species (Hymenoptera: Braconidae) previously recorded in Tasmania and included in a phylogenetic work on the braconid subfamily Rogadinae (*Quicke et al., 2021*). By investigating bycatch taxa in this manner, DNA metabarcoding provides a very powerful tool for ecological research, not only capable of identifying target and non-target species, but also of determining the diversity and distribution of taxa within a region, even if these may be new to science (*Buchner et al., 2023*).

## CONCLUSIONS

The high-throughput metabarcoding approach used in this study enabled comparisons between different trapping methods used for biosecurity and ecological surveillance, revealing differences in the number, diversity, and species composition of trapped insects, with different taxonomic groups preferentially detected by different traps. Furthermore, we found that not only the number of traps used, but their spatial distribution across crops

and proximity with the orchard's edges can influence the insect diversity captured. This highlights how accurately detecting pest species as well as obtaining reliable measurements of insect diversity both heavily depend on the trapping method used, the number of replicate traps deployed, and the spatial distribution of these traps, even when monitoring a single farm. This also underscores the importance of appropriate trap selection during the experimental design phase, always considering which trap may be better suited to capture the target species within the environment that is being sampled. Furthermore, the first record of *Casuarinicola* cf *warrigalensis* in Victoria demonstrates how non-destructive insect metabarcoding can be extremely accurate in recording potential exotic species, even when these are not expected to be found in a given location (and therefore often not targeted by conventional surveillance techniques). However, our study also highlighted the importance of comprehensive and curated DNA reference databases, the lack of which may result in mistakenly attributing a DNA sequence to the wrong species.

We recommend further collection and sequencing of poorly represented taxa, in cooperation with expert taxonomists, in order to improve taxonomic identification using metabarcoding and other molecular approaches. Until then, with the vast majority of insect groups being understudied and/or scarcely represented on public nucleotide databases, non-destructive DNA extraction methods remain of paramount importance for validating molecular results and confirming taxonomical identifications with traditional morphological examination.

## ACKNOWLEDGEMENTS

The authors would like to thank Paul McClure (SuniTAFE Smart Farm) for collecting the samples from the field. Thanks to David Madge (Agriculture Victoria) for providing technical support with the wind traps. Thanks to Gary Taylor (The University of Adelaide) for the helpful comments on the psyllid species identification. Thanks to Isabel Valenzuela-Gonzales and Mallik Malipatil (Agriculture Victoria) for providing morphological identification for some of the insect specimens. Thanks to Caitlin Selleck, James Buxton and Lixin Eow (Agriculture Victoria) for providing comments on some of the insect species identified here. Thanks to Shakira Johnson (AusVeg) for providing technical support to the iMapPESTS Sentinel system in the field.

### Funding

This work was supported by the iMapPESTS project, Horticulture Innovation Australia: ST16010, Australian Government Department of Agriculture as part of its Rural R & D for Profit program. The funders had no role in study design, data collection and analysis, decision to publish, or preparation of the manuscript.

### Grant Disclosures

The following grant information was disclosed by the authors:
Horticulture Innovation Australia: ST16010.

Australian Government Department of Agriculture as part of its Rural R & D for Profit program.

## Competing Interests

The authors declare that they have no competing interests. Jessica Lye is employed by Citrus Australia Ltd.

## Author Contributions

- Francesco Martoni conceived and designed the experiments, performed the experiments, analyzed the data, prepared figures and/or tables, authored or reviewed drafts of the article, set up experiments in the field, and approved the final draft.
- Reannon Smith performed the experiments, prepared figures and/or tables, authored or reviewed drafts of the article, and approved the final draft.
- Alexander M Piper performed the experiments, analyzed the data, authored or reviewed drafts of the article, and approved the final draft.
- Jessica Lye conceived and designed the experiments, authored or reviewed drafts of the article, set up experiments in the field, and approved the final draft.
- Conrad Trollip performed the experiments, authored or reviewed drafts of the article, and approved the final draft.
- Brendan C Rodoni conceived and designed the experiments, authored or reviewed drafts of the article, and approved the final draft.
- Mark J Blacket conceived and designed the experiments, authored or reviewed drafts of the article, and approved the final draft.

## Field Study Permissions

The following information was supplied relating to field study approvals (*i.e.*, approving body and any reference numbers):

Sampling was conducted at the SuniTAFE Smart Farm. No collection permit was required.

## DNA Deposition

The following information was supplied regarding the deposition of DNA sequences:

The raw data generated for this work is considered sensitive for biosecurity reasons and therefore not publicly available. Raw data can be made available upon request.

## Data Availability

Code are available at GitHub: https://alexpiper.github.io/iMapPESTS/local_metabarcoding.html.

The amplicon sequence variants (ASVs) generated for this study are available in the Supplemental Files.

The raw data generated in this study originates from post-border biosecurity surveillance activities and due to its sensitive nature is stored on a secure server within the Victorian Government Biosciences Advanced Scientific Computing infrastructure (BASC).

The raw sequencing data supporting the conclusions of this study can be made available upon reasonable request. Enquiries can be addressed to the AgriBio Compliance Team (agribio.compliance@agriculture.vic.gov.au), and should include the title of the manuscript and the names of the authors.

## Supplemental Information

Supplemental information for this article can be found online at http://dx.doi.org/10.7717/peerj.15831#supplemental-information.

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
