# Peer review of "Non-destructive insect metabarcoding for surveillance and biosecurity in citrus orchards: recording the good, the bad and the psyllids"

_PeerJ, doi:10.7717/peerj.15831_

## Round 0.1 · original submission · Major Revisions

Dear authors,

The three reviewers found your manuscript suitable for consideration for publication in PeerJ. Before publication, however, several questions raised by reviewers need to be addressed. Consequently, I recommend major revisions for your manuscript based on the interesting topic.

Reviewer 1 ·

Basic reporting

Sir,
I found the work 'Non-destructive insect metabarcoding for surveillance and biosecurity in citrus orchards: recording the good, the bad and the psyllids' very interesting, however the quality of MS in terms of writing part of this MS, citing related references is not up to the mark, which is a mandatory pre-requisite for publication. Keeping this in view the article maty be subject to acceptance after major revision.

Attaching the annotated pdf for the detailed comments given for the authors revision.

Experimental design

Satisfactory

Validity of the findings

Satisfactory

Additional comments

None

Annotated reviews are not available for download in order to protect the identity of reviewers who chose to remain anonymous.

Reviewer 2 ·

Basic reporting

Clear, unambiguous, professional English language used throughout. Introduction & background to show context. The results presented in the abstract need to rewrite and should be more focused along with relevant data to provide some useful information to the readers about the results of Non-destructive insect metabarcoding. Literature well referenced & relevant. The structure conforms to PeerJ standards, discipline norm. Figures are relevant, high quality, well labelled & described and properly cited in the text. Raw data supplied. However, corrections in the references style are to be done (journal issue number should be in bold style).

Experimental design

no comment

Validity of the findings

no comments

Additional comments

The article presents several important findings which enabled comparisons between different trapping methods, showing not only difference in the insect catch volumes, but also in their taxonomical assignments, with different insect groups collected by different traps. Furthermore, the impact of spatial variation on insect diversity across the same orchard. Not only the number of traps used, but their position across crops and their proximity with the orchard’s edges can impact the insect diversity assessed. This highlighted how measurements of insect diversity and biodiversity assessments in general may depend on trapping method, number of traps and position of traps, even within the same farm. Finally, the record of the new species proved how non-destructive insect metabarcoding can be extremely accurate in recording potential exotic species, even when these are not expected to be found in a given location.
Although the article title is indicating about the importance of Non-destructive insect metabarcoding for surveillance and biosecurity in citrus orchards but in the introduction, the aim of the study relevant to the article title kept at last i.e. iv explore the value of metabarcoding (line no 104), similarly in results and discussion, the authors gave priority in highlighting the different sampling methods and comparing them with respect to the diversity of insect and natural enemies. This diverts the focus of the readers from the main topic of non-destructive insect metabarcoding.

Reviewer 3 ·

Basic reporting

Comments to authors

The research article titled “Non-destructive insect metabarcoding for surveillance and biosecurity in citrus orchards: recording the good, the bad and the psyllids” deals with the comparative study of various traps across different locations in citrus orchards. The research article is well formulated, relevant and carried out intensively which will generate information and scientific insight to new species of good and bad insects in citrus crop. The study is original fitting the scope of the journal. There are several minor typing errors throughout the text. The result and discussion part consists of only results and there is absence of references supporting the claims in discussion section. Therefore, I would suggest the authors to carry out a careful and extensive revision of the text and include the required mentioned data to make the article more significant and impactful.


Abstract: Abstract should be recasted with information into materials methods and results with findings.
Introduction: Restructure the introduction part with a conceptual understanding and include more recent references pertaining to good and bad insects in citrus orchards. Information regarding highest productivity of citrus, and its area dominance needs to be indicated. Include biotic and abiotic factors affecting the HLB disease.
Material methods: Materials and methods satisfactory. Kindly include the total area coverage of the study.
Kindly justify the seasonal preference and duration taken for study.
Result and discussion: Discussion part recent references to justify findings are missing. Kindly add references.


Section Line no Comments
Introduction 36 Mention the causal organism of “HLB”
35-37 Kindly recast the line.
49 Kindly add reference
60 Together they constitute 52%. Kindly provide information for the rest 48%.
62 Indicate the reference.
68-72 Recast the line/ too lengthy sentence.
91-98 Data related to ‘material and method’. Kindly reassemble.
Materials and methods 118 Grammatical error.
133 Avoid abstract lines “previously described’. provide quantified data.
212 Explain the reason why ground and wind traps were collected weekly.
Results and Discussion 386 Kindly recast
396 Why the samples collected from wind traps (9%) and 1.51% failed library preparation.
405-422 The findings are not supported by references. Kindly justify with references.
472-490 References to support findings
Conclusion Add relevance of your study and future thrust.

Experimental design

Satisfactory

Validity of the findings

All the information are reflected and well stated.

---

## Round 0.2 · accepted · Accept

Dear Authors,

I am pleased to inform you that the reviewers have recommended your publication based on your significant improvements to the manuscript. Your efforts in addressing their comments and suggestions have been acknowledged, and the reviewers now consider the paper to be suitable for publication.

Reviewer 1 ·

Basic reporting

The authors have addressed all the comments now, the article may be accepted in its current form.

Experimental design

Satisfactory

Validity of the findings

Satisfactory

Additional comments

NA

Reviewer 3 ·

Basic reporting

The authors have revised the manuscript as per my comments. Now it can be published in its current form.

Experimental design

NA

Validity of the findings

NA

Additional comments

The authors have revised the manuscript as per my comments. Now it can be published in its current form.